# Characterization of Self-Heating Process in GaN-Based HEMTs

**Daniel Gryglewski [1] [ID], Wojciech Wojtasiak [1],* [ID], Eliana Kamińska [2],* and Anna Piotrowska [3],***

[1]   Institute of Radioelectronics and Multimedia Technology, Warsaw University of Technology, Nowowiejska 15/19, 00-662 Warsaw, Poland; dgrygle@ire.pw.edu.pl

[2]   Institute of High Pressure Physics Unipress, Al. Prymasa Tysiaclecia 98, 01-142 Warsaw, Poland

[3]   Lukasiewicz Research Network—Institute of Electron Technology, Al. Lotników 32/46, 02-668 Warsaw, Poland

*   Correspondence: wwojtas@ire.pw.edu.pl (W.W.); eliana@ite.waw.pl (E.K.); ania@ite.waw.pl (A.P.); Tel.: +48-22-234-5886 (W.W.)

**Abstract:** Thermal characterization of modern microwave power transistors such as high electron-mobility transistors based on gallium nitride (GaN-based HEMTs) is a critical challenge for the development of high-performance new generation wireless communication systems (LTE-A, 5G) and advanced radars (active electronically scanned array (AESA)). This is especially true for systems operating with variable-envelope signals where accurate determination of self-heating effects resulting from strong- and fast-changing power dissipated inside transistor is crucial. In this work, we have developed an advanced measurement system based on DeltaV$_{GS}$ method with implemented software enabling accurate determination of device channel temperature and thermal resistance. The methodology accounts for MIL-STD-750-3 standard but takes into account appropriate specific bias and timing conditions. Three types of GaN-based HEMTs were taken into consideration, namely commercially available GaN-on-SiC (CGH27015F and TGF2023-2-01) and GaN-on-Si (NPT2022) devices, as well as model GaN-on-GaN HEMT (T8). Their characteristics of thermal impedance, thermal time constants and thermal equivalent circuits were presented. Knowledge of thermal equivalent circuits and electro–thermal models can lead to improved design of GaN HEMT high-power amplifiers with account of instantaneous temperature variations for systems using variable-envelope signals. It can also expand their range of application.

**Keywords:** GaN HEMT; self-heating effect; microwave power amplifier; thermal impedance; thermal time constant; thermal equivalent circuit

## 1. Introduction

It is now commonly accepted that high electron-mobility transistors based on gallium nitride (GaN HEMT) are the best choice for high-frequency and high-power devices, such as high-power amplifiers (HPAs) used in new generation radars, including active electronically scanned array (AESA), and modern wireless communication systems, i.e., LTE-A and 5G radios [1–4]. Due to the large complexity of signals applied in modern radars and new radios, power amplifiers have to meet stringent requirements concerning not only their linearity, output power level and efficiency but also appropriate heat management [5]. This is because both kinds of mentioned systems are operated by variable-envelope signals.

In AESA radar pulses, the signals are not only frequency-modulated but can be also modulated both in phase and amplitude [6,7]. The same concerns high-speed wireless networks which use quadrature amplitude modulation (QAM) with a large peak-to-average power ratio (PAPR). For instance, the PARP

is 8.5–10 dB for LTE-A and more than 13 dB for 5G [8]. Such high variations in output power result in large changes of power dissipated in a transistor of HPA and thereby temperature variations in an active area of transistor. In addition, the transistors in transmitting amplifiers are often operated under fast-changing thermal conditions as response to quick and large signal envelope changes in time, e.g., during RF pulse or pulse-to-pulse [9], as well as the LTE-A signal.

It is obvious that temperature fluctuations cause changes in the transistor's electrical characteristics. This is a serious problem for the variable-envelope signals which are particularly sensitive to distortions caused by HPAs as a result of nonlinearity and self-heating effect of transistor. The impact both effects and discloses itself in the form of amplifier transmittance changes in time [10]. In the case of power amplifiers for wireless networks, parameters such as AM-AM and AM-PM conversions are often used to describe the effect of transistor nonlinearity. However, these are static parameters that do not show the instantaneous variations in signal amplitude and phase.

We think that the impact of transistor self-heating effect on its performances should be determined by the time-dependent transmittance changes, showing separately the changes in amplitude and phase.

Transistor nonlinearities have been quite thoroughly described and implemented in popular large-signal models, e.g., Angelov's HEMT model [11]. In contrast, the thermal effect in the transistor, especially GaN HEMT is not so widely recognized and is usually modeled using either $R_{th}$ under static conditions or a simple 1-section low-pass (parallel $R_{th}$-$C_{th}$) circuit as in the microwave transistor models provide by e.g., Wolfspeed (Durham, NC, USA), MACOM (Lowell, MA, USA), Modelithics (Tampa, FL, USA). This approach is not satisfying for fast-changing variable-envelope excitations. Therefore, we propose to describe self-heating effect in transistor by means of transient thermal impedance $Z_{th}(t)$ and more complex, multi-section, equivalent thermal circuit of transistor.

The paper presents our own approach to the thermal characterization of GaN HEMTs using thermal impedance measurements $Z_{th}(t)$ correlated with the solution of heat conduction equation in the GaN HEMT structure by means of FDTD method [12]. As a result that GaN-based epi-structures for HEMTs are grown on various substrates, namely on SiC, Si and GaN and consist of several epi-layers, each with different thermal properties. The results of simulations and measurements enable the thermal time constants appearing in the transistor structure to be identified.

## 2. Scope of the Research

### 2.1. Samples

The following GaN-based HEMTs have been a test subject in our study: commercially available GaN-on-SiC HEMTs CGH27015F (packaged) from Wolfspeed, and TGF2023-2-01 (die) from Qorvo (Greensboro, NC, USA), GaN-on-Si HEMT NPT2022 (packaged) from MACOM and GaN-on-GaN HEMT (marked T8) fabricated within PolHEMT project [13,14]. The T8 transistor was made on epitaxial layers grown on a truly bulk monocrystalline semi-insulating GaN. Testing was performed on two gates of 0.8 µm length and 500 µm width structure. The dies were mounted to the plate in test board using EPO-TEK H20E glue approx. 0.025 mm. Typical performance and electrical characteristics of the selected GaN HEMTs are given in Table 1 (manufacturer's data).

**Table 1.** Typical performance and application information of high electron-mobility transistors based on gallium nitride (GaN HEMT) samples.

| Parameter | CGH27015F | TGF2023-2-01 | NPT2022 | T8 |
|---|---|---|---|---|
| Frequency Range | 2.3–2.9 GHz | DC–14 GHz | DC–2 GHz | DC–12 GHz |
| Breakdown Voltage $V_{DS}$ | 120 V | 40 V | 160 V | 120 V |
| Channel Temperature | 175 °C | 225 °C | 200 °C | 295 °C |
| Thermal Resistance $R_{th}$ | 8 °C/W | 16 °C/W | 1.7 °C/W | 30 °C/W |
| Test Conditions ($V_{DS}$, $I_{DQ}$) | 28 V, 100 mA | 28 V, 125 mA | 48 V, 600 mA | 28 V, 65 mA |
| Saturated Output Power | 15 W | 5 W | 100 W | 2 W |
| Power Gain | 13 dB@2.5 GHz | 16 dB@3 GHz | 16 dB@2 GHz | 14 dB@3 GHz |
| View | | | | |

### 2.2. Electrical Characterization of Thermal Properties

In general terms we follow the well-known DeltaV$_{GS}$ measurement technique at the constant current of forward-biased gate-to-source diode in MESFET or HEMT (MIL-STD-750D-3 standard, method 3104) [15,16]. However, our approach is different from the previous ones by specific bias and timing conditions of the transistor during the measurements.

The knowledge of thermal impedance $Z_{th}(t)$ allows calculating the channel temperature $T_j(t)$ for any shape of dissipated power $P_d(t)$ as follows [17]:

$$T_j(t) = T_0 + \int_0^t P_d \times Z'_{th}(t - \tau)d\tau \tag{1}$$

where:

$T_j(t)$—channel temperature response;

$T_0$—ambient temperature (heatsink);

$Z'_{th}(t)$—time derivative of $Z_{th}(t)$;

$P_d(t)$—dissipated power.

The thermal impedance $Z_{th}(t)$ of different elements is often modeled, through the electro–thermal analogy, by lumped electrical equivalent circuit which contains a number of thermal resistances $R_{th}$ and thermal capacitances $C_{th}$ connected in an appropriate way. Typically, in simplified terms, the thermal equivalent circuit consists of several parallel $R_{th}$-$C_{th}$ circuits connected in series [18]. Each of the low-pass circuits $R_{thi}$-$C_{thi}$ corresponds to a thermal time constant $\tau_i$ in an exponential approximation of thermal impedance $Z_{th}(t)$ characteristic. Such a lumped electrical model can be used to calculate the channel temperature using one of popular circuit simulators like ADS or SPICE. That approach is very convenient, because the temperature of active area of transistors can be simulated using the tool applied anyway for analysis of electrical parameters.

As previously mentioned the developed system of thermal impedance $Z_{th}(t)$ measurement was inspired by the method 3104 from MIL-STD-750D-3 standard which uses the effect of the voltage drop $\Delta V_{GS}(t)$ at a forward–biased junction as a sensor of the temperature. Furthermore, the temperature values of the gate-source diode and the transistor channel are identical. The basic formula for that measurement technique was derived from the Schottky's diode equation and is expressed as follows:

$$V_{GS} = n \times V_b - \frac{n \times k \times T_j}{q}\left[2ln(T_j) - ln\left(\frac{I_G}{A \times W}\right)\right] \tag{2}$$

where:

$I_G$—junction forward current;
$A$—effective Richardson constant;
$W$—the junction surface;
$q$—charge of the electron;
$k$—Boltzmann constant;
$n$—ideality factor;
$V_b$—built-in barrier voltage.

At the constant gate current $I_G$, the voltage drop $\Delta V_{GS}$ across the source—gate junction decreases almost linearly with the rise of temperature and is given by:

$$\Delta V_{GS} = K \times \Delta T_j \tag{3}$$

where: $K$—constant.

The $Z_{th}(t)$ measurement procedure consists of the following steps:

- preparation of a test board with connected transistor marked as DUT in Figure 2b.
- gate-source voltage $V_{GS}(t)$ recording.
- $K$ factor measurement—calibration.
- thermal impedance $Z_{th}(t)$ calculation.

Examples of test boards for packaged transistors and chips measurements are presented in Figure 1a,b, respectively. The test board consists of a printed board circuits (PCB) placed on a thick metal base plate (usually made of copper) which should be characterized by high thermal inertia. The transistor bottom is thermally connected to this plate. The temperature at the bottom of the transistor ($T_0$) must be constant during the $V_{GS}(t)$ recording. Otherwise the $T_0$ temperature changes must be taken into account in the last step of the procedure i.e., thermal impedance $Z_{th}(t)$ calculation. The transistor can be biased in active state in heating phase of the $V_{GS}(t)$ recording procedure. Since the GaN HEMTs are generally potentially unstable the stabilizing circuits are required on the PCB to protect the transistor against damage.

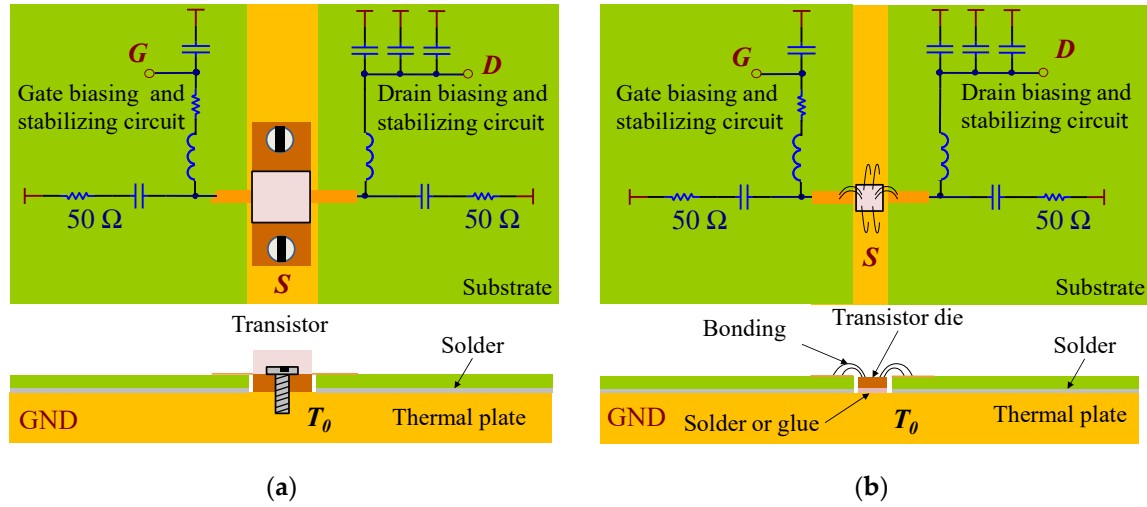

**Figure 1.** The outline of the test boards: (**a**) packaged transistor and (**b**) die (chip).

The simplified block diagram of the proposed $Z_{th}(t)$ measurement system with the timing diagram of $V_{GS}(t)$ recording procedure is shown in Figure 2. In the methods based on the MIL-STD-750D-3 standard the gate-to-source diode is forward-biased all the time and the operating point of the transistor

during the heating phase is placed in "on" region of the DC I-V output characteristic i.e., with relatively low $V_{DS}$ DC voltage ($V_{DS} \leq V_{knee}$) and high $I_D$ drain current. In that conditions the power dissipated in transistors is significantly lower and hence the $R_{th}$ value is also smaller than in the case of normal transistor operation in transmitter's amplifier when RF signal is amplified i.e., the average value of the drain current is higher than at the quiescent operating point (without RF power, as in classes AB, B and C). Therefore, in our thermal measurements, the operating point of the tested transistor is selected so that it corresponds to the expected maximum power from the amplifier, especially when typical $V_{DS}$ bias voltage for power GaN HEMTs is over a 28 V to 50 V voltage range.

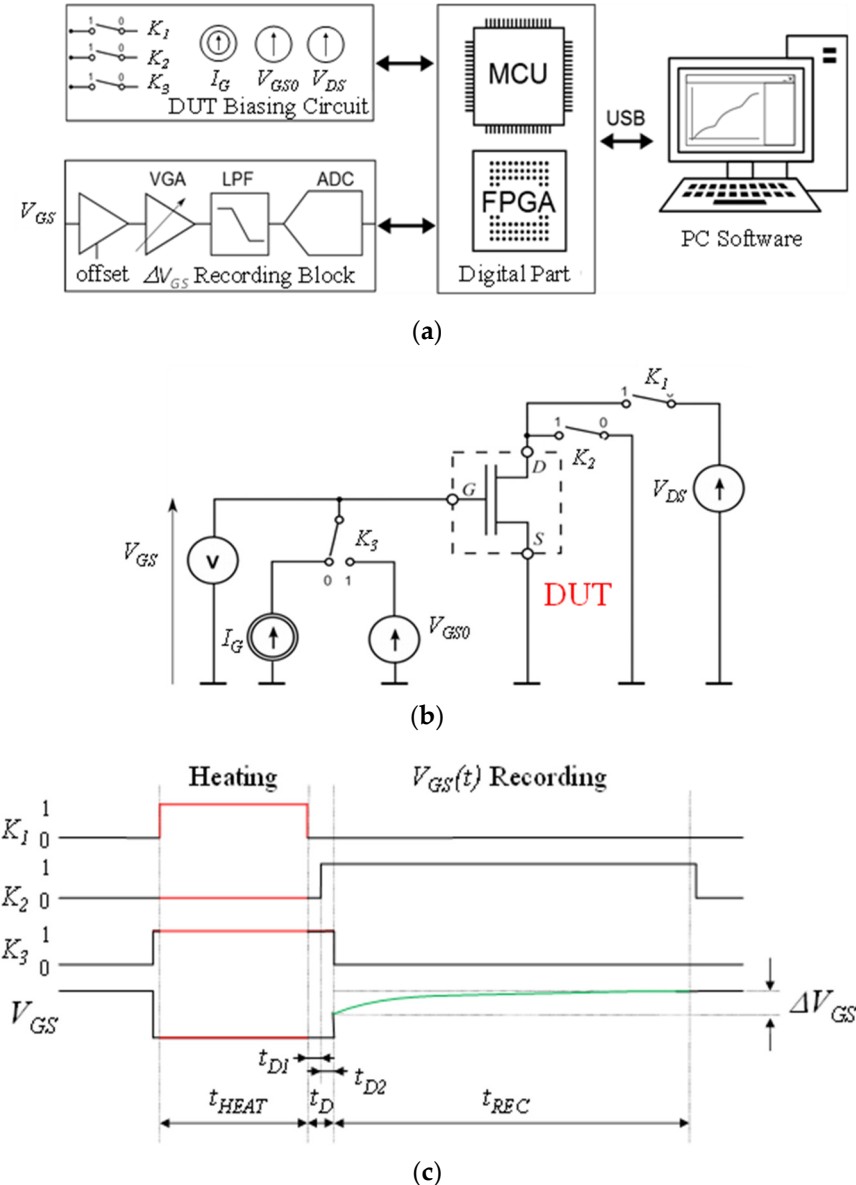

**Figure 2.** (**a**) Simplified block diagram of thermal impedance $Z_{th}(t)$ measurement system; (**b**) DUT biasing circuit; (**c**) timing diagram of $V_{GS}(t)$ recording procedure.

Information on the influence of $V_{DS}$ voltage on thermal resistance $R_{th}$ is scarce and ambiguous. From one side strong thermal resistance $R_{th}$ changes versus $V_{DS}$ voltage for GaAs MESFET transistors (the constant level of power was dissipated in transistors during the tests) was observed [15]. On the other hand weak dependence of $Z_{th}(t)$ or $R_{th}$ for GaAs (no more than 10%) [19] and GaN (no more

6%) [20,21] transistors for $V_{DS}$ change was demonstrated. As reported in [22–24], the $Z_{th}(t)$ changes mainly follow dependence of thermal conductivity of GaN HEMT layers on temperature.

Our test setup (Figure 2) was designed to be very flexible, and it allows setting wide range $V_{DS}$ bias voltages and $I_D$ current during the heating phase. It includes extra $K_2$, $K_3$ switches and controlled $V_{GS0}$ voltage source. The $V_{GS0}$ setting range is −6–0 V and $V_{DS}$ from 0 V to +50 V. The control range of current source $I_G$ is 0.1–10 mA. The proposed $V_{GS}(t)$ recording procedure (Figure 2c) is also different from the method based on MIL-STD-750D-3 standard.

At the beginning the keys are set up in the following positions: $K_1$, $K_3$ position "1", $K_2$ position "0" and the transistor is biased at the chosen operating point and heated by DC power dissipated therein. After the heating pulse the keys $K_1$, $K_2$ and $K_3$ are switched to the positions 0 and 1, respectively. This is the start of $V_{GS}(t)$ sampling during the cooling phase of the DUT. The forward gate current remains constant and it equal to $I_G$ after the heating time while the drain-source voltage source $V_{DS}$ circuit is open i.e., drain current is 0. The switching time of $K_1$, $K_2$ and $K_3$ keys is less than 10 ns but in practice, the total switching time between heating end and the start of recording phase $t_D$ is less than 100 ns. In the commercially available measurement systems that time is close to 5 µs [25]. The $t_D$ time depends on input capacitance $C_{gs}$ of transistor and $I_G$ value. Therefore, gate current $I_G$ value should be as high as possible. The forward gate current is limited by the maximum allowed level specified for the transistor. Furthermore, time delay is also determined by bias and stabilization circuits. In MIL-STD-750D standard the $V_{GS}$ voltage is only measured in two-time moments: before heating and as quick as possible after heating. These two $V_{GS}$ values allow calculating only the thermal resistance $R_{th}$. This is the main purpose of MIL-STD-750D-3 standard as it is clearly indicated in description. Therefore, this method is aimed at the testing transistors in packages, especially die attachment quality [15].

As shown in Figure 2a, the $V_{GS}(t)$ sampling is performed by means the recording block and digital part of the system. The recording time $t_{REC}$ as well as the sampling frequency $f_s$ can be changed. At the beginning of the recoding phase $f_s$ achieves 100 MHz and after 1 s drops to 10 Hz. The recording is continued up to the moment when the lack is significant changes of $V_{GS}(t)$. The maximum recording time of $V_{GS}(t)$ is 90 s. This ploy enables significantly reducing the amount of $V_{GS}(t)$ recorded data. The $V_{GS}(t)$ and $Z_{th}(t)$ characteristics are similar to the response of low pass filter. Therefore, there is no need to record of $V_{GS}(t)$ samples with the maximum sampling frequency up to end of measurement procedure.

After $V_{GS}(t)$ recording, the calibration is needed to calculate $K$ factor values. The DUT is placed in a thermal test chamber and the $V_{GS}$ voltages across the forward-biased gate-to-source Schottky junction for a number of different temperatures are stationary measured. The $I_G$ value is constant and the same as during in $V_{GS}(t)$
recording. The concept of $K$ factor measurement is presented in Figure 3.

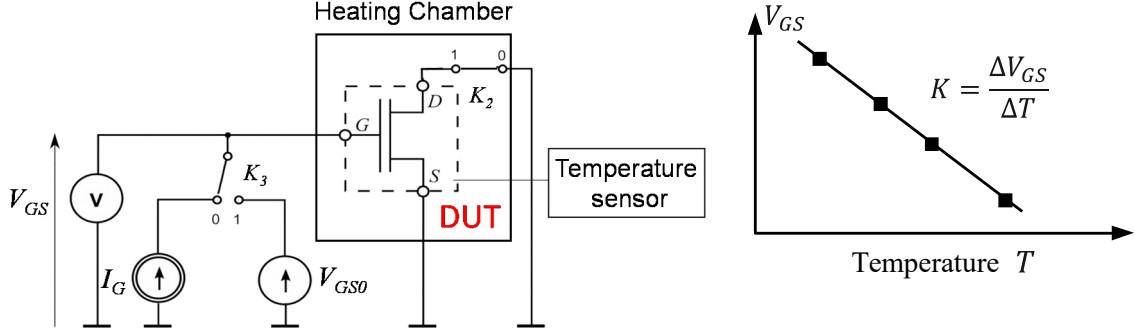

**Figure 3.** The concept of $K$ factor measurement.

The last step of the measurement procedure is channel temperature $T_j(t)$ and thermal impedance calculation $Z_{th}(t)$. The channel temperature $T_j(t)$ of transistor is given by following formula:

$$\Delta T_j(t) = T_0 + K \times V_{GS}(t) \tag{4}$$

As shown in Figure 2c the $V_{GS}(t)$ is acquired during the transistor cooling. Under these conditions, as shown in Figure 2c, the thermal impedance $Z_{th}(t)$ can be calculated as follows:

$$Z_{th}(t) = \frac{T_j(0) - T_j(t)}{P_{dc}} \tag{5}$$

where:

$T_j(0)$—calculated channel temperature at the beginning of the $V_{GS}(t)$ recording;
$P_{DC}$—dissipated power in transistor during heating phase.

The $Z_{th}(t)$ measurement system consists of the hardware (microcontroller and FPGA), firmware and PC software. The FPGA block controls the $K_1$–$K_3$ switches, acquires and stores the $V_{GS}(t)$ data from A/D converter. The communication between hardware and PC is realized by microcontroller (MCU) using USB standard. The $V_{GS}(t)$ data from FPGA is transferred to PC via MCU. The MCU controls the components of the recording block and the $V_{GSO}$, $V_{DS}$ voltage sources. All parameters of $V_{GS}(t)$ recording procedure (Figure 2b,c) can be set using PC software. The PC software also allows pre-processing of the received data, calculation of $Z_{th}(t)$ impedance and finally visualization of the results. The graphical user interface of the PC software is presented in Figure 4a. The $Z_{th}(t)$ measurement results can be exported to text file in *.csf format. The PC software has been written in Java using the Eclipse environment. The photo of hardware of the $Z_{th}(t)$ measurement system is shown in Figure 4b.

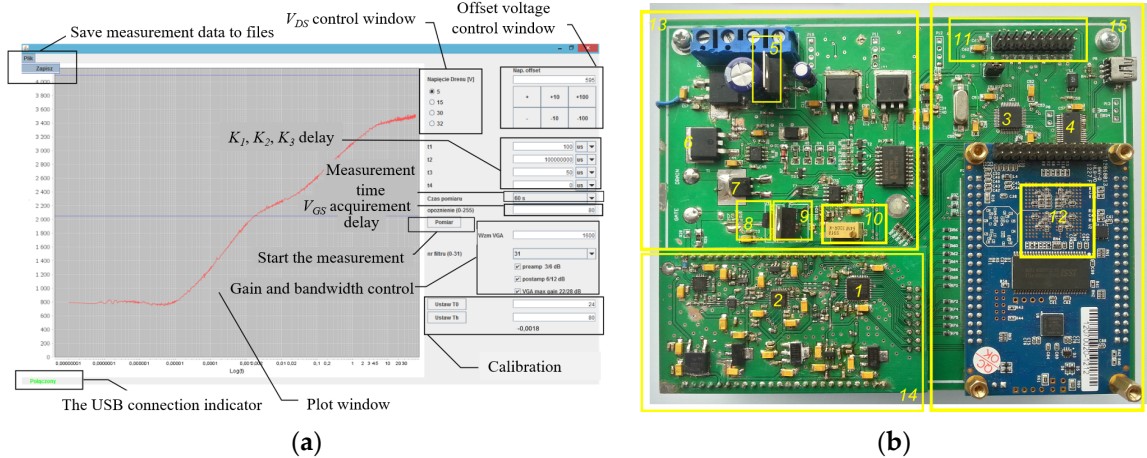

(**a**)                                                        (**b**)

**Figure 4.** (**a**) Graphical user interface (GUI); (**b**) photo of $Z_{th}(t)$ measurement system hardware. (1—A/D converter, 2—VGA, 3—MCU, 4—USART/USB converter, 5—$V_{DS}$ voltage regulator, 6—$K_1$ switch, 7—$K_2$ switch, 8—$I_G$ current source, 9—$V_{GS}$ voltage regulator, 10—$V_{GS0}$ adjusting, 11—MCU programing connector, 12—FPGA located at bottom side, 13—DUT biasing circuit, 14—recording block, 15—digital part of the system).

## 3. Results

The $Z_{th}(t)$ and $T_j(t)$ of CGH27015 and T8 are presented in Figure 5a–d, respectively. The GaN-on-SiC HEMT CGH27015F was mounted in the test board shown in Figure 1a. During the heating phase the CGH27015F was biased as follows: $V_{DS} = 28$ V ($P_D = 14$ W) and $V_{DS} = 15$ V ($P_D = 7$ W) at the same current $I_D = $ ~0.5 A.

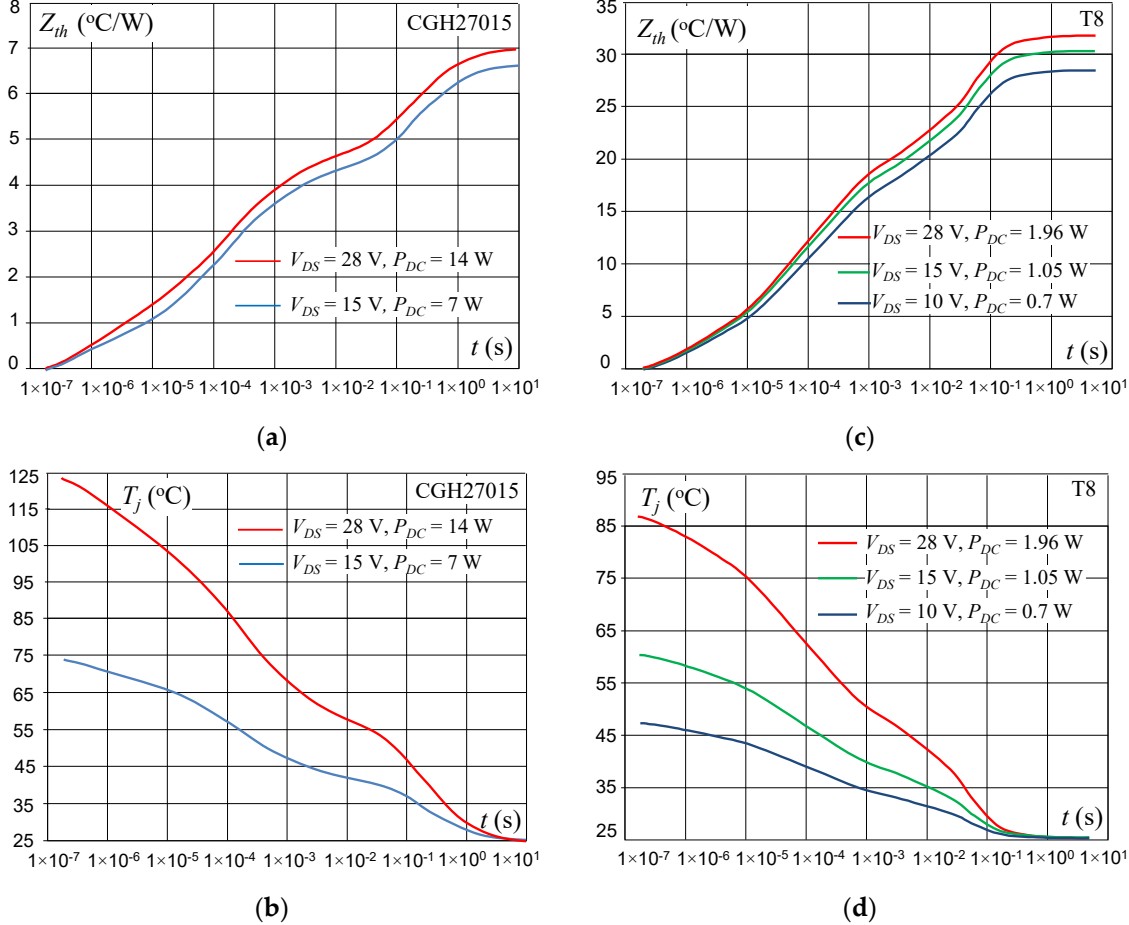

**Figure 5.** GaN-on-SiC CGH27015F and GaN-on-GaN HEMT T8 (**a,c**) thermal impedance $Z_{th}(t)$ and (**b,d**) channel temperature $T_j(t)$ measurements, respectively.

During the heating phase the T8 was biased: $V_{DS} = 28$ V ($P_D = 1.96$ W), $V_{DS} = 15$ V ($P_D = 1.05$ W) and $V_{DS} = 10$ V ($P_D = 0.7$ W) with the same drain current of 70 mA. The next results of T8 and GaN-on-SiC TGF2023-2-01 measurements presented in Figure 6 were performed under modified conditions i.e., the same power level was dissipated inside dies during heating phase of measurement procedure. In this phase, dissipated power level inside T8 was $P_D = 0.75$ W and voltage $V_{DS} = 5$ V, 10 V, 15 V, 20 V and 28 V (Figure 6a,b). The $Z_{th}(t)$ and the $T_j(t)$ of TGF2023-2-01 are shown in Figure 6c. In this case the dissipated power level was $P_D = 2.5$ W and the $V_{DS} = 5$ V, 10 V, 15 V, 20 V and 28 V.

The $Z_{th}(t)$ changes (T8) shown in Figure 5c, when different power levels were dissipated in GaN HEMTs, are bigger in comparison to $Z_{th}(t)$ changes indicated in Figure 6c for the same power dissipated in transistors and different drain-to-source voltages. The impact of Ga-on-Si HEMT NPT2022 voltage $V_{DS}$ on the $Z_{th}(t)$ characteristics, as shown Figure 7a, is slightly larger to similar $Z_{th}(t)$ characteristics of GaN-on-SiC HEMT TGF2023-2-01 (Figure 6a). Generally, obtained results confirm the lack of significant dependence of impedance $Z_{th}(t)$ on GaN HEMT bias voltage.

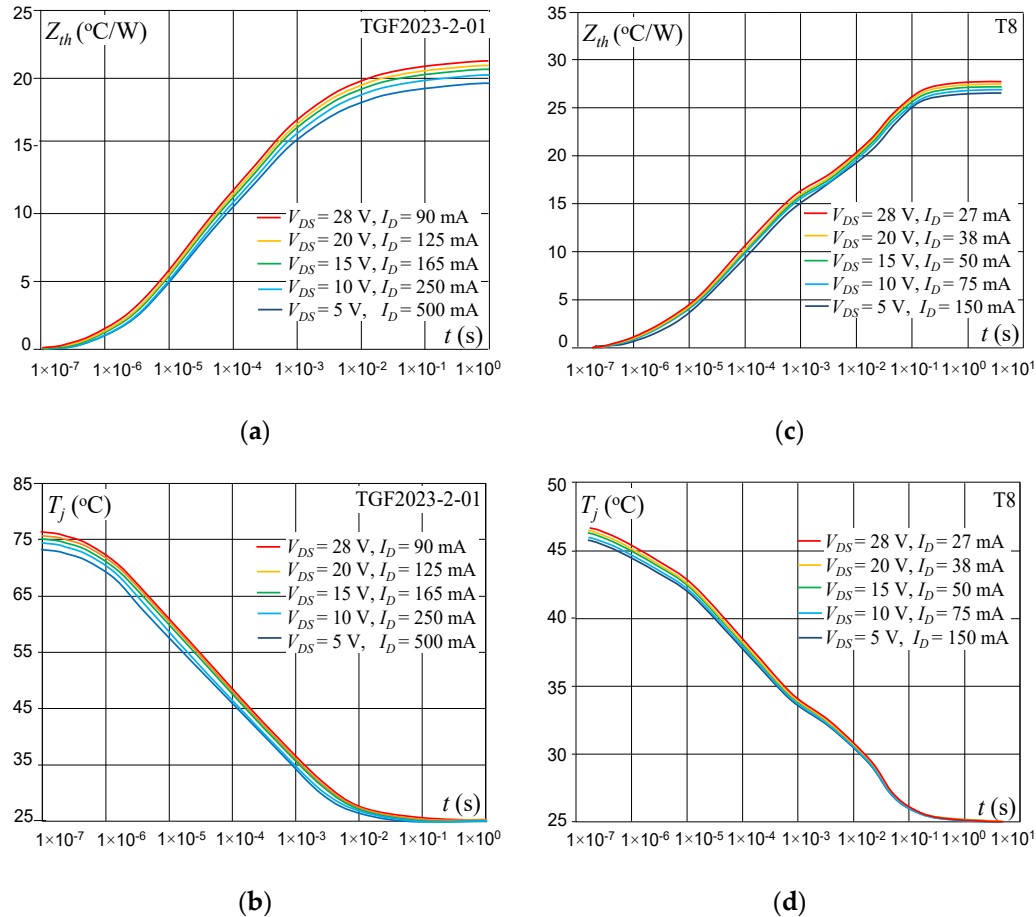

**Figure 6.** Thermal impedance $Z_{th}(t)$ and channel temperature $T_j(t)$ measurements for the same power level dissipated inside dies during the heating phase: (**a**,**b**) GaN-on-SiC TGF2023-2-01 (Qorvo), (**c**,**d**) GaN-on-GaN HEMT T8, respectively.

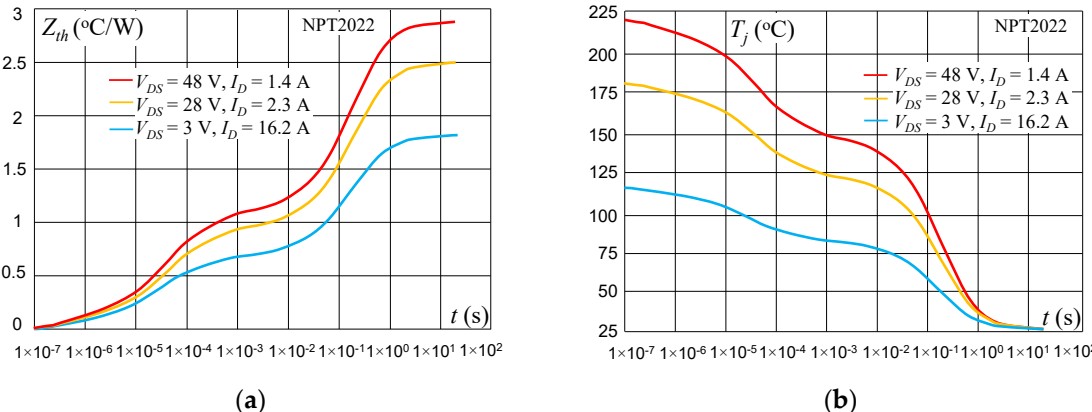

**Figure 7.** GaN-on-Si HEMT NPT2022 (MACOM) (**a**) thermal impedance $Z_{th}(t)$ and (**b**) channel temperature $T_j(t)$ measurements.

The "tank" and "filter" configurations of $R_{th}$-$C_{th}$ thermal model are considered [18]. These configurations are also known as Cauer and Foster. Both models are capable to quite accurately fit the thermal impedance $Z_{th}(t)$ characteristics. The "tank" circuit consists of a chain of parallel circuits $R_{th}$-$C_{th}$ which is simple to mathematical description. We have developed the automatic routine of "tank" model fitting in Mathcad software. The input data for this software are the $Z_{th}(t)$ measurement

results stored in text format file (*.csf). For assumed number of $R_{th}$-$C_{th}$ cells, the software allows calculating maximum error of fitting curve.

The thermal "tank" models of selected HEMTs were calculated at following bias points: CGH27015F—$V_{DS}$ = 28 V, $I_D$ = 0.5 A, T8—$V_{DS}$ = 28 V, $I_D$ = 72 mA, TGF2023-2-01—$V_{DS}$ = 28 V, $I_D$ = 90 mA, NPT2022—$V_{DS}$ = 48 V, $I_D$ = 1.5 A. These operating points correspond to output power levels close to the maximum for each transistor. The $Z_{th}(t)$ characteristics were fitted to measurements with the error lower than 1% and are shown in Figure 8.

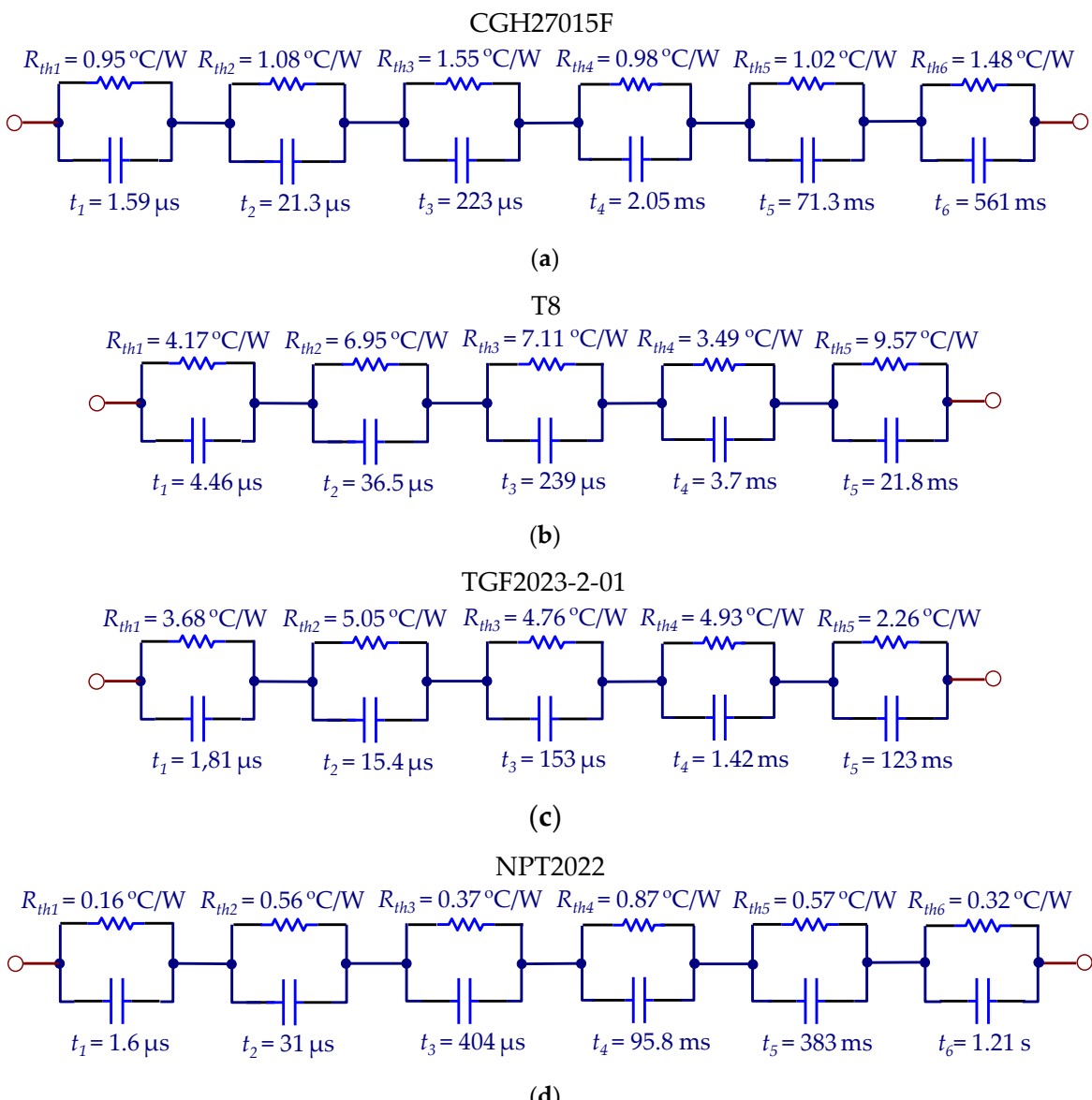

**Figure 8.** (**a**) CGH27015F (Wolspeed), (**b**) T8 PolHEMT, (**c**) TGF2023-2-01 (Qorvo), (**d**) NPT2022 (MACOM) thermal GaN HEMT models.

As shown in Figure 8, the thermal models of packaged devices CGH27015F and NPT2022 are more complicated than T8 and TGF2023-2-01 die models. The last thermal cell in CGH27015F and NPT2022 models correspond to flange (or package) and thermal attachment to the cooling plate. Thermal time constants referred to individual epi-layers of GaN HEMT and depend on the sizes and material properties. However, it is rather impossible to identify in such a way physical properties of GaN-based epi-layers as the 3-D thermal problem has been reduced to the equivalent of a lumped element.

To verify thermal impedance measurements the transistor T8 was thermal modeled using 3-dimensional equation of heat conduction which is solved by means of FDTD method [12–26]. The T8 die modeled GaN HEMT structure and the assumed heat model flow are shown in Figure 9a,b, respectively. The heating area was located under the top transistor metallization, marked "red" in Figure 9. The constant heat density across all heating areas was assumed. The thermal parameters of transistor materials were constant and temperature independent too. Their values are shown in the transistor heat model flow. To reduce the simulation time the adoptive mesh was applied. The minimal mesh size was 1 μm and it was at the top of thermal structure. The calculations were performed in MATLAB. Due to the very time-consuming calculations the thermal plate size was reduced. The simulation of T8 thermal impedance $Z_{th}(t)$ take about 24 h on PC equipped with i7 Intel processor and 16 GB RAM. The simulations and measurements of thermal impedance $Z_{th}(t)$ of GaN HEMT T8 are shown in Figure 10.

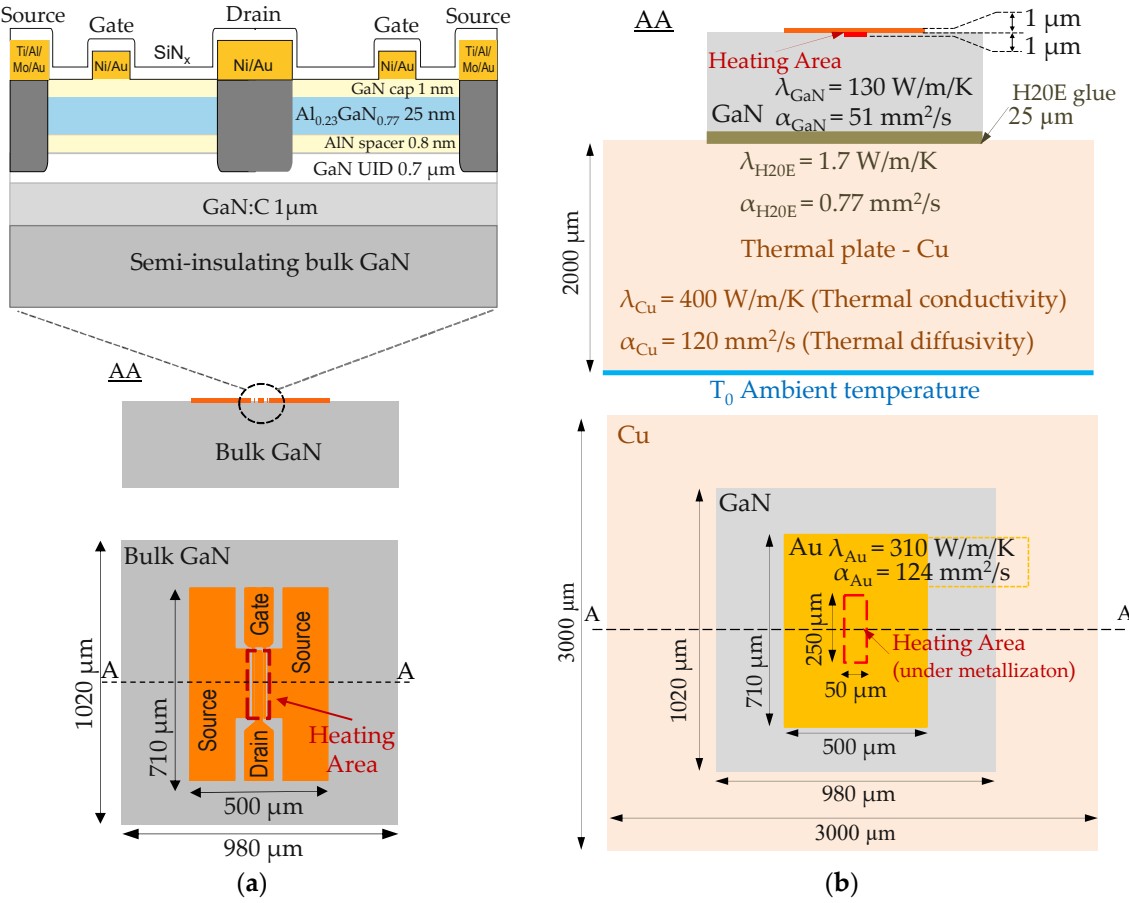

**Figure 9.** (**a**) GaN-on-GaN HEMT T8 structure; (**b**) T8 thermal model for FDTD simulation.

As shown in Figure 10, the $Z_{th}(t)$ calculations and measurements are consistent. The highest difference is at the beginning of $Z_{th}(t)$ characteristic and it is probably caused by the too large mesh of 1μm. The HEMT heating area thickness across vertical direction is much smaller.

Good compliance of the thermal measurements with the manufacturer's data was also achieved. For example, the thermal resistance measurement of CGH27015F is $Z_{th}(t \rightarrow \infty) = 7$ °C/W (Figure 5a) while $R_{th}$ value given in the datasheet is 8 °C/W in the section "absolute maximum ratings" [27]. $R_{th}$ values given by manufacturers are usually the "worst case" across the production.

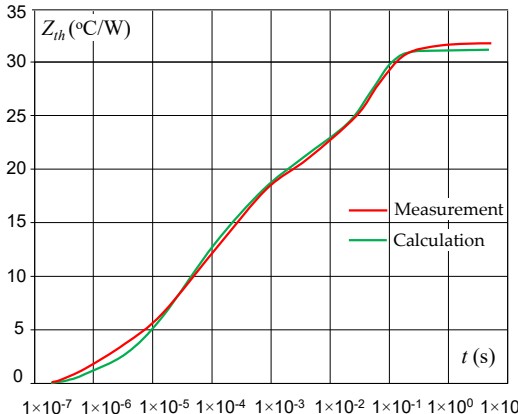

**Figure 10.** Calculated and measured thermal impedance $Z_{th}(t)$ of GaN-on-GaN HEMT T8.

Analyzing thermal characteristics (Figures 5 and 7) and thermal models shown in Figure 8a,d of packaged transistors CGH27015F and NPT2022 a significant difference in thermal resistance of GaN HEMTs on SiC and Si substrates may be observed. Considering only semiconductor structure (without flange) of both HEMTs with scaling factor ca. 5 (as output power ratio with correction for size of chip and package) GaN-on-Si HEMT NPT2022 $R_{th}$ exceeds 9 while for GaN-on-SiC HEMT $R_{th}$ is approx. 4.5 °C/W. This fact has been confirmed during measurements of the L-band 100 W amplifier with NPT2022 under pulse and CW (continuous wave) operation conditions. In case CW amplifier excitation, far below maximal output power obtained at the pulse operation, both output power and gain dropped sharply [28]. Moreover, the case temperature increased rapidly above recommended value by manufacturer. To correctly compare thermal properties of GaN-on-SiC HEMT (Quorvo TGF2023-2-01) and GaN-on-GaN (T8 HEMT) their size ought to be normalized, i.e., the size of TGF2023-2-01 scaled down to the size of T8 structure. Taking into consideration that GF2023-2-01 is ten-gates structure of 0.3 μm length and 1.25 mm width while T8 HEMT is two-gates of 0.8 μm length and 500 μm width and that the thickness of SiC substrate stands for 90% of the total thickness of GF2023-2-01 HEMT while T8 consists of lattice-matched GaN-based structure one would expect two times higher thermal resistance of GaN-on-GaN HEMT while in reality it is only 30% higher. The reason for that is boundary-effect leading to additional thermal boundary resistance at the interface of SiC substrate and GaN-based epi-structure in GaN-on-SiC HEMT [29,30].

## 4. Conclusions

Novel approach to characterizing self-heating process in GaN-based HEMTs has been proposed. It relies on measuring thermal impedance $Z_{th}(t)$ basing on MIL-STD-750D-3 standard and followed by solving 3-D heat conduction equation by means of FDTD. The thermal impedance $Z_{th}(t)$ of the GaN HEMT is calculated from the gate-to-source voltage measurements of the forward biased diode during cooling time after the heating pulse. A characteristic feature of our method is that during the heating phase the HEMT is biased at the operating point in which it will operate during its normal use in the transmitter's amplifiers of modern radar and wireless communication systems. Furthermore, the time delay between the heating end and the start of monitoring of $V_{GS}$ samples is less than 100 ns while the commercial measurement systems typically have delays as long as 5 μs. The $Z_{th}(t)$ characteristics enable the thermal time constants to be calculated.

The above procedures have been successfully applied to characterization of various commercial GaN HEMTs, namely CGH27015F (Wolfspeed) and TGF2023-2-01 (Qorvo) on SiC substrate, and NPT2022 (MACOM) on Si substrate as well as with T8 laboratory GaN-on-Ammono GaN HEMT.

The value of thermal resistance $R_{th}$ values calculated using thermal measurements for commercially available GaN-on-SiC HEMTS are consistent with manufacturer's data. The impact of material substrate on thermal features of GaN-based transistors is clearly visible. Specifically, GaN-on-Si HEMTs show

much worse thermal parameters than GaN-on-SiC. The thermal characteristics of dies i.e., TGF2023-2-01 and T8 are very similar.

The main advantage of the proposed approach is that it allows taking into account the self-heating effect of transistors during design of microwave devices. That kind of knowledge can be very important in the design of high-power amplifiers for systems using variable-envelope signals such as LTE-A and 5G radios. In addition, our method enables the thermal time constants referred to the individual GaN HEMT layers to be identified. The obtained multi-section thermal equivalent circuit of transistor and resulting thermal model may be included in GaN HEMT electrical models which are implemented in popular RF and microwave simulators. Since the GaN HEMT consists of several layers, each with different thermal properties, our measurements allow evaluating heat flow across the structure as well as determining an attachment quality die to flange or package. This is especially important when designing amplifiers with transistor chips or transistors in housing for soldering on printed board circuits (PCB).

**Author Contributions:** Conceptualization, D.G. and W.W. concept and implementation of modified DeltaVGS measurement system, E.K. and A.P.; GaN-on-SI Ammono GaN HEMTs, conceiving building blocks and process flow of device structures, methodology, D.G. and W.W.; GaN HEMT thermal impedance measurements, software, D.G. and W.W.; software development, calculation of parameters of GaN HEMT thermal equivalent circuits, simulations of GaN HEMT thermal characteristics using 3-D FDTD method, validation, D.G., W.W. and A.P.; formal analysis, W.W.; investigation, D.G., W.W., E.K. and A.P.; resources, D.G., W.W., E.K. and A.P.; original draft preparation, W.W.; manuscript review and editing, E.K. and A.P.; visualization, D.G. and W.W.; supervision, W.W. and A.P.; project administration, A.P.; funding acquisition, W.W. All authors have read and agreed to the published version of the manuscript.

**Funding:** The research was partly supported by the National Centre for Research and Development, PolHEMT Project, Contract No. PBS1/A3/9/2012 and Project "Technologies of semiconductor materials for high power and high frequency electronics" Contract No. TECHMATSTRATEG1/346922/4/NCBR/2017.

**Acknowledgments:** The authors highly appreciate valuable input to the development of GaN-based HEMTs done by Marek Ekielski, Maciej Kozubal, and Andrzej Taube from the LRN-ITE.

**Conflicts of Interest:** The authors declare no conflict of interest.

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
