# Peer review of "Characterization of Self-Heating Process in GaN-Based HEMTs"

_electronics, doi:10.3390/electronics9081305_

Round 1
Reviewer 1 Report
The authors have performed thermal analysis on several RF GaN HEMTs with different structures and packages. Both experimental and numerical simulation results were reported. In general, the data is well presented. However, there are several questions to be addressed before the manuscript can be recommended for publication.
- Fig.6, for the GaN-on-SiC TGF2023-2-01 device from Qorvo, the thermal resistance is obviously lower than that of T8 and the channel temperature drops much quicker, as the SiC substrate has a much better thermal conductivity than GaN (T8). The comment in line 286-287 sounds inappropriate.
- For the device on Si substrate from MACOM, the device seems being tested at a much higher operation power, the thermal resistance is much lower than that of the rest devices. The authors should comment on this.
- The authors should give more details on the some major parameters setting for numerical simulation.
- some typos in line 229 and other places. please double check.
Author Response
Response to Reviewer 1 Comments
We are grateful to the Reviewer for comments and suggestions that enabled us to improve significantly the manuscript.
Our revision is clearly highlighted using the "Track Changes" function in Microsoft Word, so that changes are easily visible to you.
Point 1: Fig.6, for the GaN-on-SiC TGF2023-2-01 device from Qorvo, the thermal resistance is obviously lower than that of T8 and the channel temperature drops much quicker, as the SiC substrate has a much better thermal conductivity than GaN (T8). The comment in line 286-287 sounds inappropriate.
Response 1:
There is no doubt that thermal resistance of GaN-on-SiC (Quorvo TGF2023-2-01) is smaller than that of GaN-on-GaN (T8 HEMT) and that the one of main reason behind is better conductivity of SiC substrate comparing with that of GaN. However, the difference is not as much as could be expected from simple comparison of SiC and GaN thermal conductivity. Taking into consideration the scaling effect, i.e. lateral dimensions and thickness of both HEMTs (TGF2023-2-01 is ten-gates structure of 0.3 µm length and 1.25 mm width while T8 HEMT is two-gates of 0.8 µm length and 500 µm width; the thickness of foreign SiC substrate stands for 90% of the total thickness of TGF2023-2-01 HEMT while T8 consists of lattice-matched GaN-based structure) one would expect two times better thermal conductivity of GaN-on-SiC HEMT while in reality thermal resistance of GaN-on-GaN is only 30% higher. The reason for that is boundary-effect leading to additional thermal boundary resistance at the interface of SiC substrate and GaN-based epi-structure
The comment in line 286-287 has been replaced by the following explanation.
“To correctly compare thermal properties of GaN-on-SiC HEMT (Quorvo TGF2023-2-01) and GaN-on-GaN (T8 HEMT) their size ought to be normalized, i.e. the size of TGF2023-2-01 scaled down to the size of T8 structure. Taking into consideration that GF2023-2-01 is ten-gates structure of 0.3 µm length and 1.25 mm width while T8 HEMT is two-gates of 0.8 µm length and 500 µm width and that the thickness of SiC substrate stands for 90% of the total thickness of GF2023-2-01 HEMT while T8 consists of lattice-matched GaN-based structure one would expect two times higher thermal resistance of GaN-on-GaN HEMT while in reality it is only 30% higher. The reason for that is boundary-effect leading to additional thermal boundary resistance at the interface of SiC substrate and GaN-based epi-structure in GaN-on-SiC HEMT”.
Point 2: For the device on Si substrate from MACOM, the device seems being tested at a much higher operation power, the thermal resistance is much lower than that of the rest devices. The authors should comment on this.
Response 2: GaN-on-Si HEMT NPT2022 achieves more than 100W of output power in L-band. There were two reasons for testing of NPT2022 with high dissipated power level. The first is because the measurements of high-power transistors should be performed nearby maximal output power i.e. under recommended operating conditions. The second is because thermal resistance Rth depends on temperature difference channel-to-case Tj-c, generated by the dissipated power, its level should correspond to the output power close to the maximum value i.e. under recommended operating conditions of this transistor. Indeed, the thermal resistance of NPT2022 is much lower than that of the other devices under investigation because it is high-power HEMT. Again for appropriate comparison the size of GaN-on-Si HEMT NPT2022 was scaled down to the GaN-on-SiC HEMT CGH27015F and differences in output power was taken into consideration as well. Consequently, it turned out that thermal resistance Rth of scaled NPT 2020 is approx. 2-times higher than of GaN-on-SiC HEMT CGH27015F.
Point 3: The authors should give more details on the some major parameters setting for numerical simulation.
Response 3: All thermal model parameters of T8 transistor used in our simulations are given in Fig.9(b). The minimum mesh size was 1µm on the top of thermal structure. The maximum 32 µm mesh size was applied to simulate on the bottom of copper thermal plate. Simulations were performed with minimum time step of 5 ns. The dissipated power level was 750mW. We developed own-made software for solving of 3D heat conduction equation using “direct” FDTD method in multilayer thermal structure as shown in Fig.9b. Our program in MATLAB code contains about 1600 lines.
Point 4: some typos in line 229 and other places. please double check.
Response 3: Typos and errors in the numbering of Figures have been corrected.
Line 227 - Figure 5a-> Figure 5c
Line 228 - Figure 6-> Figure 6c
Line 230 - Figure 7-> Figure 7a

Reviewer 2 Report
Tye paper "Characterization of Self-Heating Process in GaN-2based HEMTs" is an important study. The paper is well written and explained nicely.
Few comments:
The ideality factor in equation 2 has temperature dependency?
Minor correction:Change symbol ÷ to - for range (Ex. -6V - 0V), several places
All Figure: Explanation missing. Why there is impedance steps? And why same thing in Tj vs t plots?
Finally more references are required to support experimental data (So, please add some references in the experimental data analysis.).
Otherwise paper is Ok.
Author Response
Response to Reviewer 2 Comments
Dear Reviewer,
We would like to thank you for your comments and suggestions that helped us to improve our manuscript.
Our revision is clearly highlighted using the "Track Changes" function in Microsoft Word, so that changes are easily visible to you.
Point 1: The ideality factor in equation 2 has temperature dependency?
Response 1: An ideality factor "n" is a fitting parameter to achieve a compliance of a measured I(V) diode characteristic with the "ideal" Schottky equation. The experimentally reported dependence of the ideality factor on temperature in GaN-based Schottky diodes for the temperature range RT-400 K is weak [1-3]. The ideality factor decreases with increasing temperature due to the dominant thermionic emission conduction transport mechanism with increasing temperature. As explained in lines 184-187 and Figure 3, in the calibration phase, measured characteristics VGS(T) at IG=const of several different transistors were linear in a desired temperature range and under assumed bias conditions IG.
[1] Sadoun, A.; Mansouri, S.; Chellali, M.; Lakhdar, N.; Hima, A.; Benamara, Z. Investigation, analysis and comparison of current-voltage characteristics for Au/Ni/GaN Schottky structure using I-V-T. Materials Science-Poland 2019, 37, p. 496-502.
[2] Helal, H, Benamara, Z, Arbia, MB, et al. A study of current‐voltage and capacitance‐voltage characteristics of Au/n‐GaAs and Au/GaN/n‐GaAs Schottky diodes in wide temperature range. Int J Numer Model El. 2020; 33:e2714. https://doi.org/10.1002/jnm.2714
[3] Ravinandan, M.; Koteswara Rao P.; Rajagopal Reddy, V. Analysis of the current–voltage characteristics of the Pd/Au Schottky structure on n-type GaN in a wide temperature range, Semiconductor Science and Technology, Vol. 24, No 3, 2 February 2009, IOP Publishing Ltd
Point 2: Minor correction: Change symbol ÷ to - for range (Ex. -6V - 0V), several places.
Response 2: As rightly pointed out, the corrections have been introduced.
Point 3: All Figure: Explanation missing. Why there is impedance steps? And why same thing in Tj vs t plots?
Response 3: The steps of thermal impedance Zth(t) and temperature T(t) graphs result from heat flow through vertically non-homogeneous structure of HEMT. Each step in these graphs corresponds, in simplifying, heat flow across a boundary two adjacent layers in the transistor. From the point of view of mathematical description, the experimental data are fitted to a curve created by a sum of exponential functions as follows: Zth(t)=∑Ai[1-exp(t/τthi)], where: τthi = RthiCthi – thermal time constant. This is briefly mentioned on lines 235-241 (“The “tankt” and “filter”….”). Calculation of thermal time constants in the exponential approximation of Zth(t) characteristic enables problems with heat flow through transistor structure to be identified. Because the thermal time constants correspond to the individual layers of GaN HEMT and are dependent on the thickness and physical properties of each layer, the research for determining of the exact relationships between the thermal time constants and the layer parameters is currently led.
Point 3: Finally more references are required to support experimental data (So, please add some references in the experimental data analysis.).
Response 3: We have added more references relating to the experimental data analysis: [26, 28-30]).
